

# A sequential recommendation method using contrastive learning and Wasserstein self-attention mechanism

Shengbin Liang[1], Jinfeng Ma[1], Qiuchen Zhao[2], Tingting Chen[1], Xixi Lu[1], Shuanglong Ren[1], Chenyang Zhao[1], Lei Fu[3] and Huichao Ding[4]

[1] School of Software, Henan University, Kaifeng, Henan, China
[2] Shandong Jining Tobacco Co., Ltd, Jining, Shandong, China
[3] China National Tobacco Corporation Shandong Province Company, Jinan, Shandong, China
[4] Nanyang Radio and Television Station, Nanyang, Henan, China

Corresponding author
Shengbin Liang,
liangsbin@henu.edu.cn

## ABSTRACT

Recent research has demonstrated the effectiveness of utilizing contrastive learning for training Transformer-based sequence encoders in sequential recommendation tasks. Items are represented using vectors and the relations between items are measured by the dot product self-attention, the feature representation in sequential recommendation can be enhanced. However, in real-world scenarios, user behavior sequences are unpredictable, and the limitations of dot product-based approaches hinder the complete capture of collaborative transferability. Moreover, the Bayesian personalized ranking (BPR) loss function, commonly utilized in recommendation systems, lacks constraints when considering positive and negative sampled items, potentially leading to suboptimal optimization outcomes. This presents a complex challenge that needs to be addressed. To tackle these issues, this article proposes a novel method involving stochastic self-attention. This article introduces uncertainty into the proposed model by utilizing elliptical Gaussian distribution controlled by mean and covariance vector to explain the unpredictability of items. At the same time, the proposed model combines a Wasserstein self-attention module to compute the positional relationships between items within a sequence in order to effectively incorporate uncertainty into the training process. The Wasserstein self-attention mechanism satisfies the triangular inequality and can not only addresses uncertainty but also promote collaborative transfer learning. Furthermore, embedding a stochastic Gaussian distribution into each item will bring additional uncertainty into the proposed model. Multi-pair contrastive learning relies on high-quality positive samples, and the proposed model combines the cloze task mask and dropout mask mechanisms to generate high-quality positive samples. It demonstrates superior performance and adaptability compared to traditional single-pair contrastive learning methods. Additionally, a dynamic loss reweighting strategy is introduced to balance the cloze task loss and the contrastive loss effectively. We conduct experiments and the results show that the proposed model outperforms the state-of-the-art models, especially on cold start items. For each metric, the hit ratio (HR) and normalized discounted cumulative gain (NDCG) on the Beauty dataset improved by an average of 1.3% and 10.27%, respectively; on the Toys dataset improved by an average of 8.24% and 5.89%, respectively; on the ML-1M dataset improved by an average of 68.62% and 8.22%, respectively; and on the ML-100M dataset improved by

an average of 93.57% and 44.87% Our code is available at DOI: 10.5281/zenodo.13634624.

# INTRODUCTION

Sequential recommendation (SR) analyses a series of user behavior and item interactions within a specific time interval to recommend relevant items to users (*Vasile, Smirnova & Paramita, 2020*). This technique is commonly applied in e-commerce platforms (*Zhang et al., 2019*), where the user's historical behavior over a specific period is used to predict the next possible action that the user will take. SR dynamically captures users' short-term and long-term interests, thereby enabling real-time recommendations, and enhancing the accuracy of recommendation services. The essence of SR lies in effectively mining sequential relationships, with common methodologies including sequence pattern mining (*Zaki, 2020*), latent factor representation (*Rendle et al., 2019*), Markov chain-based modeling (*Rendle, Freudenthaler & Schmidt-Thieme, 2010*; *He & McAuley, 2016*), and deep learning-driven approaches (*Zhang et al., 2019*; *Wu et al., 2019*). Sequence pattern mining requires substantial computational resources and intricate rule designs, making it challenging to model complex sequences, which results in unsatisfactory recommendation accuracy. Time is a crucial factor in recommendation models, as it tracks user interest evolution; however, applying implicit factor models to implicit feedback recommendations often faces issues such as insufficient positive feedback and noisy negative feedback, thereby diminishing recommendation quality (*Cheng et al., 2021*). Markov chain modeling involves calculating the probability distribution of state transitions based on user behaviors at different moment, thereby offering insights into the conditional likelihood of behaviors occurring compared to the previous step. Nevertheless, this method still struggles with data sparsity and fails to address the long-tail effect that arises in e-commerce platforms (*Yu et al., 2019*). These approaches have inherent limitations in modeling complex features, feature interactions, and representation learning. With advancements in neural network technologies, there has been a surge of deep learning-based SR methods, including convolutional neural networks (*Gehring et al., 2017*), long short-term memory (*Hidasi, 2016*), and attention mechanisms (*Kang & McAuley, 2018*). Various SR categories exist, such as multi-behavior sequence recommendation (*Xia et al., 2022*), multi-interest sequential recommendation (*Li et al., 2019*), and multi-scale sequential recommendation (*Liu et al., 2019*, *2021*).

Contrastive learning is a technique that focuses on learning data representations by maximizing similarities between relevant samples while minimizing similarities between irrelevant samples. This approach has gained significant traction within the realm of unsupervised learning due to its flexible definition of positive and negative samples and its

high performance. Drawing inspiration from contrastive learning, researchers have integrated this approach into the domain of sequential recommendation, leading to several notable research outcomes. *Xie et al. (2022)* proposed a multi-task framework called Contrastive Learning for Sequential Recommendation (CL4SRec), aiming to extract user patterns and effective user representations. By leveraging three data augmentation methods to create self-supervised signals, the framework demonstrated better performance. User-item interactions are driven by diverse intentions, but revealing these underlying motivations can be challenging. *Chen et al. (2022)* proposed intent contrastive learning method that learns the distribution of user intentions from unlabelled user behavior sequences and incorporates the learned intentions into the sequence recommendation model, thereby improving the quality of recommendations. *Qiu et al. (2022)* proposed the DuoRec model, reshaping the distribution of sequence representations in recommendation tasks, the core idea is to measure the similarity between sequence representations and item embeddings in the shared space using dot product. *Wang et al. (2022)* presented a multi-level contrastive learning framework, MCLSR, which learns user and item representations from various perspectives at interest and feature levels through cross-view contrastive learning, resulting in enhanced recommendation performance.

Addressing data sparsity is crucial for recommendation performance, and utilizing data augmentation method plays an important role in addressing this issue. However, such methods may introduce noise. To mitigate this, *Li et al. (2023)* proposed a multi-intent oriented comparative learning recommendation framework that combines sequential patterns and self-supervised signals at the intent layer to create high quality views. *Wei et al. (2023)* developed the MoCo4SRec model, employing contrastive self-supervised learning and Momentum Contrast (MoCo) to handle sparse and noisy data effectively. Furthermore, *Du et al. (2022)* proposed a contrastive learning framework for a bidirectional Transformer called CBiT for sequential recommendation. The framework utilized a sliding window technique to divide long user sequences into finer granularities for data augmentation. By generating high-quality positive samples and employing contrastive learning, CBiT yielded improved results in sequential recommendation across four datasets.

In summary, existing contrastive learning methods in sequential recommendation typically focus on enhancing user-item interaction sequences at the data level through operations like item cropping, masking, and reordering, or by integrating auxiliary information such as multi-level and multi-intention aspects. However, these methods often struggle to provide consistent semantic enhancement samples. Various sequence models (*Kang & McAuley, 2018*; *Liu et al., 2021*; *Qiu et al., 2022*; *Sun et al., 2019*; *Xu et al., 2021*) leverage Transformer as a sequence encoder to capture item relationships and derive quality sequence representations using a self-attentive mechanism. To address these limitations, this article introduces a novel sequential recommendation model called Contrastive Learning with Wasserstein Self-Attention (CLWSR). The contributions are summarized as follows.

(1) We construct a sequential recommendation model based on contrastive learning, combined stochastic embedding to evaluate the basic interests and interest changes inherent in user behavior to improve semantic representation. In addition, employing additional regularization to constrain the distance between positive and negative sampled items to reduce the BPR loss in sequential recommendation.

(2) We propose a Wasserstein self-attention mechanism based on the Wasserstein distance to quantify differences between items within a sequence under uncertainty. The mechanism effectively addresses cold-start by facilitating collaborative transferability.

(3) We employ the cloze task masking and dropout masking, which are data-enhancing operations that generate positive samples and extend single-pair contrastive learning to multi-pair instances. In addition, we introduce a novel dynamic loss reweighting strategy to enhance the smoothing of multi-pair contrast loss.

(4) We conduct a series of experiments and ablation studies on the Beauty, Toys, ML-1M and ML-100M datasets to validate the effectiveness of CLWSR. The experimental results show that CLWSR outperforms the state-of-the-art models and achieve excellent results in both HR and NDCG.

## RELATED WORKS

### Contrastive learning

Contrastive learning is a unique unsupervised learning technique that aims to derive data representations by maximizing the similarity between related samples while minimizing the similarity between unrelated samples. Therefore, the crucial aspect of contrastive learning lies in establishing rules for generating positive and negative samples with high flexibility and customization. The basic framework of contrastive learning is showed in Fig. 1.

The input comprises three parts: positive pare $I^+$, anchor $I^G$, and negative pare $I^-$. The encoder typically consists of two sections: the backbone and the head, the head is crucial in boosting the model's performance, while the backbone is typically an encoder such as ResNet-50 or Transformer, utilized for feature extraction. The loss function can be computed as Eq. (1).

$$L_{(q,k^+,\{k^-\})} = -\log\left(\frac{\exp\left(\frac{qk^+}{\tau}\right)}{\exp\left(\frac{qk^+}{\tau}\right) + \sum_{k^-}\exp\left(\frac{qk^-}{\tau}\right)}\right). \tag{1}$$

Here q represents a query representation, $k^+$ denotes the representation of positive samples, and $\{k^-\}$ represents representation of the negative samples. $\tau$ represents a temperature hyperparameter.

BYOL is a self-supervised approach for image learning representations (*Grill et al., 2020*). It utilized online and target networks for interactive and collaborative learn. The online network is trained on augmented views of the image to predict the representation of the image seen by different augmentations in the target networks. InforNCE is a

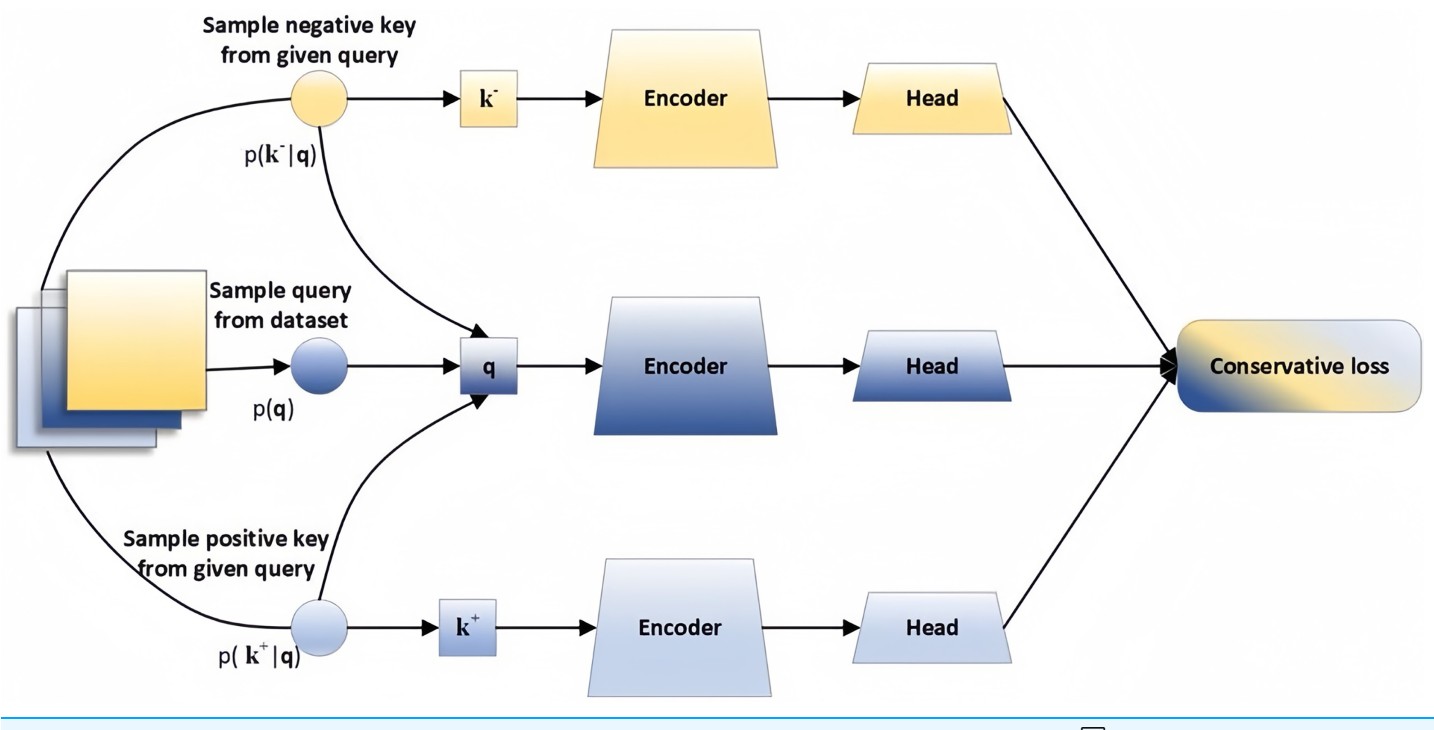

**Figure 1 Contrastive learning framework.**

discriminative approach to contrastive learning that utilized contrast prediction coding (*van den Oord, Li & Vinyals, 2018*). It focused on extracting valuable representations by leveraging probabilistic contrast loss to encourage the capture of informative spatial data relevant for predicting future samples. Negative sampling was also utilized in this method to facilitate model training. SimCLR introduced a learnable transformation between representations and contrastive loss functions to enhance the quality of learned visual representations (*Chen et al., 2020*). The framework aimed to improve the effectiveness of learned representations through contrastive losses. SwAV leveraged contrastive methods for data clustering. It incorporated a swapped prediction mechanism where the code of one view was predicted based on the representation of another view (*Caron et al., 2020*). This approach enhanced representation learning through contrastive techniques. MoCo is a method designed for unsupervised visual representation learning. MoCo had demonstrated performance comparable to supervised representation learning in various visual tasks (*He et al., 2020a*). The approach incorporated momentum contrast to enhance the quality of learned visual representations in an unsupervised setting.

The recommendation system requires full use of the user's behavior sequence to gain insight into the user's behavior and potential interests. Transformer is used to fully explore the user's behavior sequence and realize the modeling of the user's behavior sequence. For example, models such as CBiT use multi-head attention in the Transformer layer to learn the correlation between the target item and the items in the user's behavior sequence (*Du et al., 2022*). At the same time, the recommendation system needs to consider many attributes, and the graph representation can represent non-Euclidean data. The edges and

nodes of the graph can represent rich attribute information, and the graph neural network can continuously aggregate and update features to obtain the information of high-order neighbors, thereby obtaining more accurate recommendation services. LightGCN learns user and item embeddings by linearly propagating them on the user-item interaction graph, and uses the weighted sum of the embeddings learned at all layers as the final embedding (*He et al., 2020b*). Although GNN can capture high-order correlation information, it also has problems such as over-smoothing.

## Metric learning and distribution representations

Metric learning is a technique relies on a distance metric that aims to determine the similarity or dissimilarity between objects. Its goal is to bring similar objects closer together while pushing dissimilar objects further apart. Common examples of distances include the Euclidean distance, the Mahalanobis distance (*McLachlan, 1999*) and the graph distance (*Gao et al., 2010*). Distances and dot product are distinguished by their metric properties, with distances typically satisfying triangular inequalities. The triangular inequality serves as an inductive bias for distances and proves beneficial in dealing with data sparsity issues. Early works on metric learning in recommendation systems include CML (*Hsieh et al., 2017*), which utilized hinge loss to minimize the L2 distance between a user embedding and an interaction term. LRML expanded on CML by recognizing its geometric constraints and enhancing distance computation through the introduction of latent relations as translation vectors (*Tay, Anh Tuan & Hui, 2018*). TransRec drew inspiration from knowledge embedding and introduced translation vectors for sequential recommendations (*He, Kang & McAuley, 2017*). SML is a metric learning recommendation method that extended CML by incorporating additional item-centered metrics and adaptive margins (*Li et al., 2020*). Distributional representations for objects (*e.g.*, words, nodes, items) had received extensive attention in research (*Sun et al., 2018*; *Vilnis & McCallum, 2014*). Distributional representations introduce uncertainty and offer more flexibility than fixed embeddings. DVNE employed Gaussian distributions as node embeddings and proposed a deep variational model for propagating high-order neighboring information (*Zhu et al., 2018*). TIGER represented words as Gaussian distributions and introduced Gaussian attention to improve the modeling of word entailment relationships (*Vilnis & McCallum, 2014*; *Qian et al., 2021*). DDN represented users and items using Gaussian distributions and learned mean and covariance embeddings *via* a neural network (*Zheng et al., 2019*). DT4SR used distributions to represent items and learned mean and covariance *via* separate Transformers (*Fan et al., 2021*). The Wasserstein distance measured the distance between discrete and continuous distributions, enabling continuous transformation from one distribution to another while preserving distributional features. *Fan et al. (2022)* designed a Wasserstein self-attention module to capture positional relationships between items within a sequence, incorporating uncertainty into model training effectively. During the sampling process, the Wasserstein reservoir operates on sessions with higher Wasserstein distance in recommendation results, leading to lower recommendation likelihood for sessions with higher Wasserstein distance.

## PRELIMINARIES

### Problem definition

Given a set of users $U$ and items $V$ along with their interactions, we can represent user interaction sequence as $S^u = [v_1^u, v_2^u, \ldots, v_{|S^u|}^u]$, where $v_i^u \in V$ denotes the $i - th$ item interaction sequence, and $S^u$ represents the user's interaction sequence. The objective of SR is to recommend a list of top N items as the potential the next item in the sequence. Specifically, we aim to predict $p\left(v_{(|S^u|+1)}^{(u)} = v_i|S^u\right)$, which calculates the probability that the user $u$ will interact with item $v$ at the next timestamp $|S^u| + 1$.

### Self-attention for recommendation

The core component of the proposed model is the self-attention mechanism as a sequence encoder. Firstly, when provided with the users' sequence of operations $S^u$ and the maximum sequence length $n$, we truncate the sequence by removing the earliest item if $|S^u| > n$ or we pad it with zeros to create a sequence of fixed length $\mathbf{s} = (s_1, s_2, \ldots, s_n)$. Secondly, we define the item embedding matrix $\mathbf{M} \in \mathbb{R}^{|V| \times d}$, where $d$ represents the number of dimensions. Lastly, we incorporate the trainable position embedding $\mathbf{P} \in \mathbb{R}^{n \times d}$ into the sequence embedding matrix, as illustrated in Eq. (2):

$$\hat{E}_{S^u} = [m_{s_1} + p_{s_1}, m_{s_2} + p_{s_2}, \ldots, m_{s_n} + p_{s_n}]. \tag{2}$$

Specifically, self-attention is used to compute the dot product between items in a sequence as a means of determining the correlation between them, as shown in Eq. (3):

$$SA(Q, K, V) = \text{softmax}\left(\frac{QK^T}{\sqrt{d}}\right)V \tag{3}$$

where $Q = \hat{E}_{S^u}W^Q$, $K = \hat{E}_{S^u}W^K$, and $V = \hat{E}_{S^u}W^V$. Since $Q$ and $K$ use the same sequence of inputs, the scaled dot product component can learn potential correlations between items.

## METHODS

The CBiT model employed a Transformer-based sequence encoder to address the issue of semantic enhancement consistency in sequences, demonstrating good performance in sequential recommendations (*Du et al., 2022*). Our model is inspired by CBiT and utilizes a stochastic embedding layer, Wasserstein self-attention mechanism, and Transformer as the sequence encoder to generate hidden representations of the sequences. A basic linear network is used as the prediction layer to convert the hidden sequence representations into a probability distribution of candidate items.

We describe the architecture of CLWSR in this section. CLWSR incorporates contrastive learning for sequential recommendation using Wasserstein self-attention and Transformer. The structure of CLWSR is illustrated in Fig. 2. Initially, the user sequence $s_u$ generates $m$ different mask sequences. These stochastic embeddings are further transformed into embedding vectors. Subsequently, the Wasserstein self-attention module infers stochastic embeddings of mask sequences, in principle using the Wasserstein

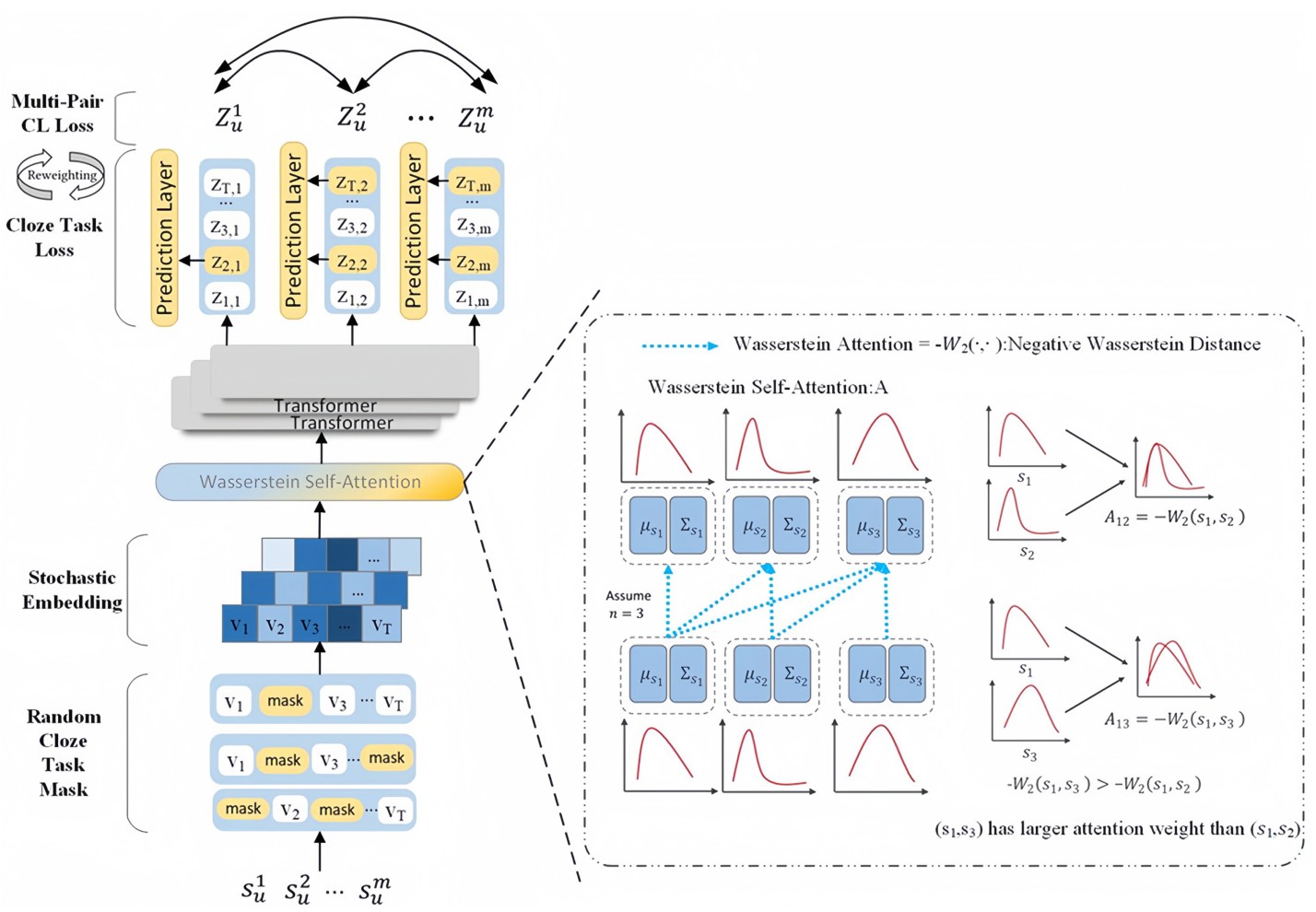

**Figure 2 The architecture of CLWSR.**

distance. The Wasserstein distance serves to quantify the distinctions between items in a sequence amidst uncertain signals. The sequences, after passing through the Wasserstein self-attention module, are subsequently fed through the Transformer, leading to the generation of a hidden representation as the final output of the last layer. Let $Z_u^1, Z_u^2, \ldots, Z_u^m$ be recognized as positive samples of the same sequence $s_u$, and let $T$ represent a collection of sequences with a maximum length. The primary objective of training is established through the cloze task, where the model is tasked with reconstructing the masked items based on their respective hidden representations. In addition, the positive sample set for multi-pair contrastive learning is the hidden representation of all masked sequences. To balance the cloze task loss and the multi-pair contrastive learning loss, CLWSR proposes a dynamic loss reweighting strategy.

The workflow of the CLWSR model includes five steps, as detailed below:

Step 1: Generate sequences with various masks derived from user operation sequences.

Step 2: Encode uncertain items by employing a multidimensional elliptical Gaussian distribution.

**Figure 3** **The workflow of CLWSR.**

Step 3: Utilize the Wasserstein distance to gauge the stochastic embedding disparity between two items and depict sequence dynamics *via* the Wasserstein self-attention mechanism.

Step 4: Transmit sequences through the Transformer to capture intricate relationships *via* non-linear processes.

Step 5: Implement contrastive learning to enhance the diversity of positive and negative samples.

The comprehensive workflow of CLWSR is shown in Fig. 3.

## Stochastic embedding layer

We depict items as distributions and incorporate uncertainty into the item embedding. The various deterministic vectors symbolize the items envisioned as random distributions, enabling a broader coverage of space and inclusion of more collaborating neighbors. We utilize a multidimensional elliptical Gaussian distribution to portray the items, with the mean and covariance vectors governing the distribution's elliptical shape. The covariance factor introduces potential uncertainty into the items being represented. We establish a mean embedding vector $M^\mu \in \mathbb{R}^{|V| \times d}$ and a covariance embedding vector $M^\Sigma \in \mathbb{R}^{|V| \times d}$ for all items. Given that the mean and covariance convey distinct signals, we introduce positional embeddings $p^\mu \in \mathbb{R}^{n \times d}$ and $p^\Sigma \in \mathbb{R}^{n \times d}$ for the mean and covariance, respectively. Consequently, for user $u$, the mean and covariance sequential embeddings can be represented as indicated in Eqs. (4) and (5).

$$\hat{E}^\mu_{s^u} = \left[ \hat{E}^\mu_{s_1}, \hat{E}^\mu_{s_2}, \ldots, \hat{E}^\mu_{s_n} \right] = \left[ m^\mu_{s_1} + P^\mu_{s_1}, m^\mu_{s_2} + P^\mu_{s_2}, \ldots, m^\mu_{s_n} + P^\mu_{s_n} \right] \qquad (4)$$

$$\hat{E}^\Sigma_{s^u} = \left[ \hat{E}^\Sigma_{s_1}, \hat{E}^\Sigma_{s_2}, \ldots, \hat{E}^\Sigma_{s_n} \right] = \left[ m^\Sigma_{s_1} + P^\Sigma_{s_1}, m^\Sigma_{s_2} + P^\Sigma_{s_2}, \ldots, m^\Sigma_{s_n} + P^\Sigma_{s_n} \right]. \qquad (5)$$

As an illustration, for the sequence's initial term $s_1$, its stochastic embedding is a Gaussian distribution of $d$-dimensional $N = \left( \mu_{s_1}, \sum_{s_1} \right)$, where $\mu_{s_1} = \hat{E}^\mu_{s_1}$ and $\sum_{s_1} = \text{diag}(\hat{E}^\Sigma_{s_1}) \in \mathbb{R}^{d \times d}$.

## Wasserstein self-attention layer

Challenges persist in modeling the dynamics of sequence information *via* stochastic embeddings. One such challenge involves efficiently depicting items with distributions while adhering to triangular inequalities. Another obstacle pertains to aggregating sequence signals effectively to derive suitable sequence representations. To tackle these hurdles, CLWSR employs the Wasserstein distance to gauge the variability among items and introduces a novel Wasserstein self-attention layer. This layer characterizes attention weights as the Wasserstein distance between items and utilizes the linear summation trait

of the Gaussian distribution to consolidate past items for acquiring a sequence representation.

### Wasserstein attention

In conventional attention mechanisms, attention weights are typically determined through similarity measures or the dot product of key-value pairs. However, this method may face challenges when processing lengthy sequences, as it necessitates calculating attention weights for all potential positions. Consequently, this results in heightened computational complexity and poses challenges in capturing long-range dependencies.

Wasserstein self-attention provides a more flexible and accurate method by using Wasserstein distance instead of the traditional similarity measure. Wasserstein distance considers the difference between the two distributions and aligns them in the different distribution, and therefore better captures the relationship between the sequences, especially when dealing with long sequences. We propose a novel self-attention variant that adapts to stochastic embeddings. The self-attention value is represented as $A \in \mathbb{R}^{n \times n}$. $A_{kt}$ signifies the attention value between item $s_k$ and item $s_t$ in the $k$-th and $t$-th positions in the sequence, where $k \leq t$ with causality considerations, respectively. According to Eq. (6), the traditional self-focused attention weight is calculated using the Eq. (6).

$$A_{kt} = \frac{Q_k K_t^T}{\sqrt{d}}. \tag{6}$$

Nevertheless, the dot product is not intended for assessing the difference between distributions (*i.e.*, stochastic embeddings) and does not uphold the triangular inequality. Therefore, the distance between the stochastic embeddings of the two items is measured using the Wasserstein distance. Formally, given two items $s_k$ and $s_t$, the corresponding stochastic embeddings are $N(\mu_{s_k}, \sum_{s_k})$ and $N(\mu_{s_t}, \sum_{s_t})$, where $\mu_{s_k} = \hat{E}_{s_k}^{\mu} W_K^{\mu}$, $\sum_{s_k} = \text{ELU}\left(\text{diag}\left(\hat{E}_{s_k}^{\Sigma} W_K^{\Sigma}\right)\right) + 1$, $\mu_{s_t} = \hat{E}_{s_t}^{\mu} W_Q^{\mu}$, $\sum_{s_t} = \text{ELU}\left(\text{diag}\left(\hat{E}_{s_t}^{\Sigma} W_Q^{\Sigma}\right)\right) + 1$.

Exponential linear unit (ELU) is used to ensure the positive definite property of the covariance, which maps inputs to the range $[-1, +\infty)$. We use the negative 2-Wasserstein distance $W_2(.,.)$ to define the attention weights, which are computed as in Eq. (7):

$$A_{kt} = -W_2(s_k, s_t) = -\left(||\mu_{s_k} - \mu_{s_t}||_2^2 + \text{trace}\left(\Sigma_{s_k} + \Sigma_{s_t} - 2\left(\Sigma_{s_k}^{\frac{1}{2}} \Sigma_{s_k} \Sigma_{s_t}^{\frac{1}{2}}\right)^{\frac{1}{2}}\right)\right). \tag{7}$$

Wasserstein distance offers several advantages. Firstly, Wasserstein distance can assess the item dissimilarity using uncertainty information because it is adept at gauging the disparity between distributions. Secondly, Wasserstein distance adheres to the triangle inequality, enabling inductive reasoning to capture collaborative transferability in sequence modelling. Lastly, while dealing with non-overlapping distributions, the Wasserstein distance provides a smoother measure, thus contributing to a more stable training process. In contrast, Kullback–Leibler divergence may yield an infinite distance, leading to numerical instability.

### Wasserstein attentive aggregation

For each position in the sequence, the similarity to other positions is first calculated using Wasserstein distance, then these similarities are converted into weights, and finally the embeddings from the previous steps are weighted and summed with these weights to obtain the output embedding for the current position. Thus, when the attention weights are normalized, weighted summation of the embeddings means that the embeddings from the previous steps are weighted and summed using these normalized weights as coefficients to obtain the output embedding for the current position. With this approach, the model can concentrate on the information in the sequence that is most relevant to the current location, leading to better modelling and prediction.

The output embedding of the item at each position in the sequence is a weighted sum of the embeddings from the previous steps, so the weighted sum of the embeddings is weighted by normalizing the attention weights, where the weights are the normalized attention values $\tilde{A}$ as in Eq. (8):

$$\tilde{A}_{kt} = \frac{A_{kt}}{\sum_{j=1}^{t} A_{jt}}. \tag{8}$$

As a result of representing each item as a stochastic embedding with both mean and covariance characteristics, these two components must be aggregated separately. We use the linear combination property of the Gaussian distribution as Eq. (9).

$$z_{s_t}^{\mu} = \sum_{k=1}^{t} \tilde{A}_{kt} V_{s_k}^{\mu}, \quad z_{s_t}^{\Sigma} = \sum_{k=1}^{t} \tilde{A}_{kt}^2 V_{s_k}^{\Sigma}. \tag{9}$$

### Transformer layer

Each Transformer block includes a multi-head self-attention module and a feed-forward network. The multi-head self-attention module for efficient access to information in various subspaces at different positions. The calculation method is shown in Eq. (10).

$$z_{s_t}^* = \frac{Q_k K_t^T}{\sqrt{d}}. \tag{10}$$

where $Q = (z_{s_t}^* W)^Q$, $K = (z_{s_t}^* W)^K$, $V = (z_{s_t}^* W)^V$, and * can be either $\mu$ or $\Sigma$.

As multi-head self-attention primarily relies on linear projections, incorporating a feed-forward network post the attention layer aids in capturing non-linear features. The calculation process is expressed as Eqs. (11) and (12).

$$FFN^{\mu}(Z_{s_t}^{\mu}) = \text{ELU}(Z_{s_t}^{\mu} W_1^{\mu} + b_1^{\mu}) W_2^{\mu} + b_2^{\mu}. \tag{11}$$

$$FFN^{\Sigma}(Z_{s_t}^{\Sigma}) = \text{ELU}(Z_{s_t}^{\Sigma} W_1^{\Sigma} + b_1^{\Sigma}) W_2^{\Sigma} + b_2^{\Sigma}. \tag{12}$$

where $W_1^* \in \mathbb{R}^{d \times d}$, $W_2^* \in \mathbb{R}^{d \times d}$, $b_1^* \in \mathbb{R}^d$, and $b_2^* \in \mathbb{R}^d$ are learnable parameters, and * can be $\mu$ or $\Sigma$. Given the numerical stability benefit of the Exponential Linear Unit (ELU), we opt for ELU over Rectified Linear Unit (ReLU). Additionally, we incorporate other

components like residual connections, normalization layer, and residual layers. The output of the layer can be expressed as:

$$Z_{s_t}^{\mu} = z_{s_t}^{\mu} + \text{Dropout}(FFN^{\mu}(\text{LayerNorm}(z_{s_t}^{\mu}))) \tag{13}$$

$$Z_{s_t}^{\Sigma} = \text{ELU}\left(z_{s_t}^{\Sigma} + \text{Dropout}\left(FFN^{\Sigma}\left(\text{LayerNorm}\left(z_{s_t}^{\Sigma}\right)\right)\right)\right) + 1. \tag{14}$$

The proposed model utilizes the combination of ELU activation and covariance embedding to ensure the positive definite nature of the covariance.

### Prediction layer

Considering the final output of any hidden representation $z_t$ at position $t$, where the mean embedding and covariance embedding at each position are summed to obtain $z_t$, we adopt a straightforward linear layer to transform $z_t$ into a probability distribution over candidate items:

$$P(v) = w^p z_t + b^p. \tag{15}$$

In Eq. (15), $w^p \in \mathbb{R}^{|V| \times d}$ is the weight matrix, and $b^p \in \mathbb{R}^{|V|}$ is the bias term for the prediction layer.

Based on our practical observations, we do not use feed forward networks with item embedding matrices (*Sun et al., 2019*), but instead choose to use basic linear layers. We found that employing a prediction layer with a shared item embedding matrix can undermine the contrastive learning task, which depends on the shared item embedding matrix to compute item similarities. In addition to reducing computational overhead, the prediction layer without the item embedding matrix also decouples the dependency between the cloze task and the contrastive learning task, making the two tasks independent of each other.

### Learning with the cloze task

The traditional BPR loss function is used to optimize the ranking of candidate items that users are interested in. It is trained by maximizing the rating difference between positive and negative samples, and is defined as shown in Eq. (16). However, this method is prone to overfitting or local optimization, resulting in uneven distribution of positive and negative samples, or too small interval between positive and negative samples.

$$L = -\sum_{(u,i,j) \in D} \log \sigma(\hat{r}_{ui} - \hat{r}_{uj}) \tag{16}$$

where $(u, i, j)$ denotes a triad where $u$ is the user, $i$ is a positive sample (items that the user likes), and $j$ is a negative sample (items that the user does not like). $\hat{r}_{ui}$ is the predicted rating of item $i$ by user $u$. $\hat{r}_{uj}$ is the predicted rating of item $j$ by user $u$. $\sigma$ is the sigmoid function used to map the rating difference between $(0, 1)$.

In this article, we generate multiple positive samples by introducing the cloze task, and in this way extend the sample generation method of traditional BPR. Unlike the traditional approach of considering only a pair of positive and negative samples for training, the new

approach allows a user to learn by comparing and contrasting with multiple items, improving the model's learning ability. In addition to this, dropout mask is a regularization technique commonly used in neural network training to avoid overfitting by randomly dropping some neurons. Here, dropout mask is not only used for neural network training but also used to generate multiple positive samples. This allows the model to generate multiple sample pairs for comparative learning during training, thus enhancing the model's learning of potential preferences and features.

In summary, to train the Transformer, the cloze task is brought in. For each iteration step, given the sequence $s_u$, we use different random seeds to generate $m$ masking sequences $s_u^1, s_u^2, \ldots, s_u^m$. In each masking sequence $s_u^j$ ($1 \le j \le m$), the proportion $\rho$ of all items in the sequence $s_u$ is randomly replaced with a masking marker (mask), and the position index of the masked item is denoted as $I_u^j$. The model requires reconstruction of the masked items. Based on top of the BPR loss function formulation, the training target of the loss function with the cloze task is defined as Eq. (17):

$$L_{\mathrm{main}} = -\sum_{j=1}^{m} \sum_{t \in I_u^j} \left[ \log \sigma(P(v_t \mid s_u^j)) + \sum_{v_t^- \notin s_u} \log 1 - \sigma(P(v_t^- \mid s_u^j)) \right]. \tag{17}$$

Here, $\sigma$ represents the sigmoid function, and the probability $P(\cdot)$ is defined as Eq. (15). Each ground truth item $v_t$ is paired with a randomly sampled negative item $v_t^-$. Note that in computing the loss function for the cloze task, we only consider masked items.

## Multi-pair contrastive learning

The purpose of contrastive learning is to bring positive samples closer to each other while separating negative samples from positive ones. Normally, given a batch of sequences $\{s_u\}_{u=1}^N$ with batch size $N$, a pair of hidden representations $Z_u^x$ and $Z_u^y$ stemming from the same original sequence $s_u$ are brought together as a pair of positive samples while the other $2(N-1)$ hidden representations from the same batch are considered negative samples (*Chen et al., 2020*). We define the contrastive learning loss for one pair based on InfoNCE (*van den Oord, Li & Vinyals, 2018*) as Eq. (18):

$$l(Z_u^x, Z_u^y) = -\log \frac{e^{\langle Z_u^x, Z_u^y \rangle / \tau}}{e^{\langle Z_u^x, Z_u^y \rangle / \tau} + \sum_{k=1, k \neq u}^N \sum_{c \in \{x,y\}} e^{\langle Z_u^x, Z_k^c \rangle / \tau}}. \tag{18}$$

where $\tau$ is a temperature hyper-parameter, and $x$ and $y$ denote the index of two different mask sequences, with $1 \le x, y \le m$. The cosine similarity function $\langle \phi_1, \phi_2 \rangle = \frac{\phi_1^T \cdot \phi_2}{||\phi_1|| \, ||\phi_2||}$ is used to calculate the similarity between two hidden representations.

## RESULTS

### Datasets

We conduct experiments on four benchmark datasets: Beauty, Toys, MovieLens-1M, and MovieLens-100M. The Amazon dataset (*McAuley et al., 2015*) contains reviews of products from various domains with relatively short sequence lengths. We choose Beauty

**Table 1 Statistics of datasets.**

| Datasets | #Users | #Items | #Actions | Avg. length | Sparsity |
|----------|--------|--------|----------|-------------|----------|
| Beauty | 22,363 | 12,101 | 198,502 | 8.9 | 99.93% |
| Toys | 19,412 | 11,924 | 167,597 | 8.6 | 99.95% |
| ML-1M | 6,040 | 3,953 | 1,000,209 | 163.5 | 95.21% |
| ML-100M | 330,975 | 86,000 | 33,000,000 | 99.8 | 99.88% |

and Toys as two distinct datasets derived from the Amazon dataset in our experiments. MovieLens-1M (ML-1M) and MovieLens-100M (ML-100M) comprise user ratings of movies with considerably long sequences (*Harper & Konstan, 2015*). All interactions are treated as implicit feedback. We eliminate duplicate interactions and compile user sequences by organizing each user's interactions chronologically. Following the data filtering techniques outlined in previous studies (*Liu et al., 2021*; *Qiu et al., 2022*; *Sun et al., 2019*), we exclude users with fewer than five interactions and items associated with fewer than five users. Our approach employs a leave-one-out evaluation setup: the last item is reserved for testing, the penultimate item for validation, and the remaining items for training. The statistical details of the preprocessed datasets are presented in Table 1.

As can be seen from Table 1, these four datasets include different purposes, the data is extremely sparse, and the average length and data size are also significantly different, especially ML-100M, whose actions scale reaches 33,000,000, and the data volume is extremely large.

## Metrics

For a fair comparison, we rank the predictions for the entire item set (*Krichene & Rendle, 2020*). We report top-K hit rate (HR@K) and NDCG@K as metrics.

$$HR@K = \frac{1}{N} \sum_{i=1}^{N} \text{hits}(i) \tag{19}$$

where $N$ is the total number of users. $\text{hits}(i)$ represents whether the value visited by the $i$-th user is in the recommendation list. It is 1 if yes, 0 otherwise.

$$NDCG@K = \frac{1}{N} \sum_{j=1}^{N} \frac{1}{\log_2(p_i + 1)} \tag{20}$$

where $p_i$ is the real visit value of the $i$-th user in the position of the recommendation list. If the value does not exist in the recommendation list, then $p_i \rightarrow \infty$.

## Hyperparameter settings

The experimental setup utilizes the PyTorch framework, featuring two Transformer blocks with two attention heads, a hidden dimension, and a batch size of 256 each. The mask ratio $\rho$ is configured at 0.15, following the recommendation from BERT. We employ the Adam optimizer (*Diederik, 2014*) with a learning rate of 0.001, $\beta_1 = 0.9$ and $\beta_2 = 0.999$, while

incorporating exponential decay of the learning rate after every 100 epochs (*Loshchilov, 2017*). In the process of hyperparameter tuning, we explore the dropout ratio from 0.1 to 0.9, the factor $\lambda$ from 1 to 9, $\tau$ from 0.1 to 6, the count of positive samples $M$ from 2 to 8, and $\alpha$ values $\{0.0001, 0.0005, 0.001, 0.05, 0.1\}$. The model is trained for 250 epochs, and checkpoints demonstrating the best NDCG@10 on the validation set are selected for testing.

## Experimental results

The presented baselines used for comparison are as follows:

- CoSeRec (*Liu et al., 2021*): Enhances CL4SRec by integrating data augmentation techniques.

- CBiT (*Fan et al., 2022*): Utilizes a bidirectional Transformer to enhance performance in sequential recommendation tasks through contrastive learning.

- HGN (*Ma, Kang & Liu, 2019*): Combines BPR to get both long-term and short-term user preferences.

- LightGCN (*He et al., 2020b*): Includes GCN-neighborhood aggregation, a pivotal component for collaborative filtering.

- TGT (*Xia et al., 2022*): Presents a temporal graph Transformer recommendation framework to capture dynamic short-term and long-term user-item interaction patterns jointly.

- HAM (*Peng et al., 2021*): Generates sequential recommendations by considering long-term preferences of users, recent activities, and item synergies. Item representation is achieved through simple pooling, while synergy modeling involves element-wise product operations.

- MADGA (*Wang et al., 2024*): MADGA dynamically transforms subsequences into graphs to capture the evolving interdependencies. Uniquely, this GA approach involves explicit alignment of both nodes and edges, employing Wasserstein distance for nodes and Gromov-Wasserstein distance for edges.

- SwiMDiff (*Tian et al., 2024*): SwiMDiff employs a scene-wide matching approach that effectively recalibrates labels to recognize data from the same scene as false negatives. This adjustment makes CL more applicable to the nuances of remote sensing.

CLWSR's performance is evaluated against these baseline models, and the experimental results are summarized in Table 2.

From the analysis presented in Table 2, the CLWSR model performs relatively stably on all four datasets and outperforms other models overall, especially on the ML-1M and ML-100M dataset, where they perform better that other baseline models. This success can be attributed to the longer average sequence length of this database compared to other datasets. The Wasserstein self-attention mechanism implemented in the model effectively captures the dependencies between various positions in the extended session sequences. In addition, leveraging the cloze task for Transformer training contributes to improve the performance of the model.

However, it can be seen on the Beauty dataset that it slightly lags behind HAM at HR@5 and MADGA at NDCG@5 and NDCG@10, respectively. From the Toys dataset, it can be

**Table 2 Experimental results.**

| Dataset | Metric | HGN | CoSeRec | LightGCN | TGT | HAM | CBiT | MADGA | SwiMDiff | CLWSR |
|---------|--------|-----|---------|----------|-----|-----|------|-------|----------|-------|
| Beauty | HR@5 | 0.0276 | 0.0504 | 0.0285 | 0.0653 | 0.0736 | 0.0637 | 0.0684 | 0.0588 | 0.0732 |
| | HR@10 | 0.0459 | 0.0726 | 0.0853 | 0.0835 | 0.0982 | 0.0905 | 0.0987 | 0.0892 | 0.1125 |
| | HR@20 | 0.0788 | 0.1035 | 0.1156 | 0.1088 | 0.1199 | 0.1223 | 0.1193 | 0.1082 | 0.1233 |
| | NDCG@5 | 0.0332 | 0.0339 | 0.0174 | 0.0491 | 0.0421 | 0.0451 | 0.0592 | 0.0391 | 0.0520 |
| | NDCG@10 | 0.0421 | 0.0410 | 0.0231 | 0.0529 | 0.0506 | 0.0537 | 0.0679 | 0.0599 | 0.0589 |
| | NDCG@20 | 0.0655 | 0.0488 | 0.0538 | 0.0623 | 0.0671 | 0.0617 | 0.0770 | 0.0606 | 0.0772 |
| Toys | HR@5 | 0.0483 | 0.0533 | 0.0266 | 0.0502 | 0.0075 | 0.0640 | 0.0591 | 0.0610 | 0.0691 |
| | HR@10 | 0.0659 | 0.0755 | 0.0508 | 0.0800 | 0.1070 | 0.0865 | 0.0926 | 0.0796 | 0.1110 |
| | HR@20 | 0.0877 | 0.1037 | 0.0799 | 0.1187 | 0.1237 | 0.1167 | 0.1233 | 0.1022 | 0.1251 |
| | NDCG@5 | 0.0311 | 0.0370 | 0.0173 | 0.0378 | 0.0368 | 0.0462 | 0.0367 | 0.0568 | 0.0491 |
| | NDCG@10 | 0.0581 | 0.0442 | 0.0479 | 0.0511 | 0.0471 | 0.0535 | 0.0488 | 0.0599 | 0.0629 |
| | NDCG@20 | 0.0652 | 0.0513 | 0.0591 | 0.0692 | 0.0582 | 0.0610 | 0.0915 | 0.0703 | 0.0714 |
| ML-1M | HR@5 | 0.1231 | 0.1128 | 0.0360 | 0.0199 | 0.0098 | 0.2095 | 0.3152 | 0.2135 | 0.3648 |
| | HR@10 | 0.2064 | 0.1861 | 0.0431 | 0.0871 | 0.1730 | 0.3013 | 0.4012 | 0.2987 | 0.4091 |
| | HR@20 | 0.3081 | 0.2950 | 0.0659 | 0.1188 | 0.2142 | 0.3998 | 0.4871 | 0.3826 | 0.5243 |
| | NDCG@5 | 0.1065 | 0.0692 | 0.0368 | 0.0621 | 0.1070 | 0.1436 | 0.1312 | 0.1363 | 0.1662 |
| | NDCG@10 | 0.1449 | 0.0915 | 0.0692 | 0.0885 | 0.1387 | 0.1694 | 0.1574 | 0.1704 | 0.1728 |
| | NDCG@20 | 0.1688 | 0.1247 | 0.0749 | 0.1076 | 0.1639 | 0.1957 | 0.1892 | 0.1993 | 0.2092 |
| ML-100M | HR@5 | 0.1033 | 0.0964 | 0.0211 | 0.0136 | 0.0097 | 0.1871 | 0.1960 | 0.1899 | 0.2158 |
| | HR@10 | 0.1102 | 0.0932 | 0.0234 | 0.0198 | 0.0802 | 0.2151 | 0.1977 | 0.1900 | 0.2196 |
| | HR@20 | 0.1107 | 0.0993 | 0.0227 | 0.0211 | 0.0844 | 0.2312 | 0.2034 | 0.1972 | 0.3134 |
| | NDCG@5 | 0.0931 | 0.0764 | 0.0301 | 0.0545 | 0.1101 | 0.1456 | 0.1163 | 0.1143 | 0.1516 |
| | NDCG@10 | 0.1002 | 0.0723 | 0.0318 | 0.0691 | 0.1265 | 0.1504 | 0.1193 | 0.1159 | 0.1577 |
| | NDCG@20 | 0.1145 | 0.0788 | 0.0412 | 0.0785 | 0.1290 | 0.1610 | 0.1354 | 0.1290 | 0.1611 |

seen that it lags behind MADGA at NDCG@5 and SwiMDiff at NDCG@20, respectively. The main reason for the lagging behind is due to the fact that the three models have stronger adaptations on the shorter-length sequences, while our model is more adapted to longer length sequences. The performance enhancement observed in CLWSR model can be attributed to the synergistic influence of three key components: distributed embedding, Wasserstein self-attention, and the advantages derived from multi-pair comparison learning. The variation curves of NDCG@10 and HR@10 on four different datasets are shown in Figs. 4–7. We can see that the use of distributed embedding, Wasserstein self-attention and multi-pair comparison learning in CLWSR significantly improves performance on all four datasets.

## Model complexity analysis

The computational complexity of CLWSR is dominated by the randomized mask operation, the Wasserstein self-attention layer, the feedforward network, Transformer layer, and contrastive learning module. The Wasserstein self-attention defined in Eq. (7)

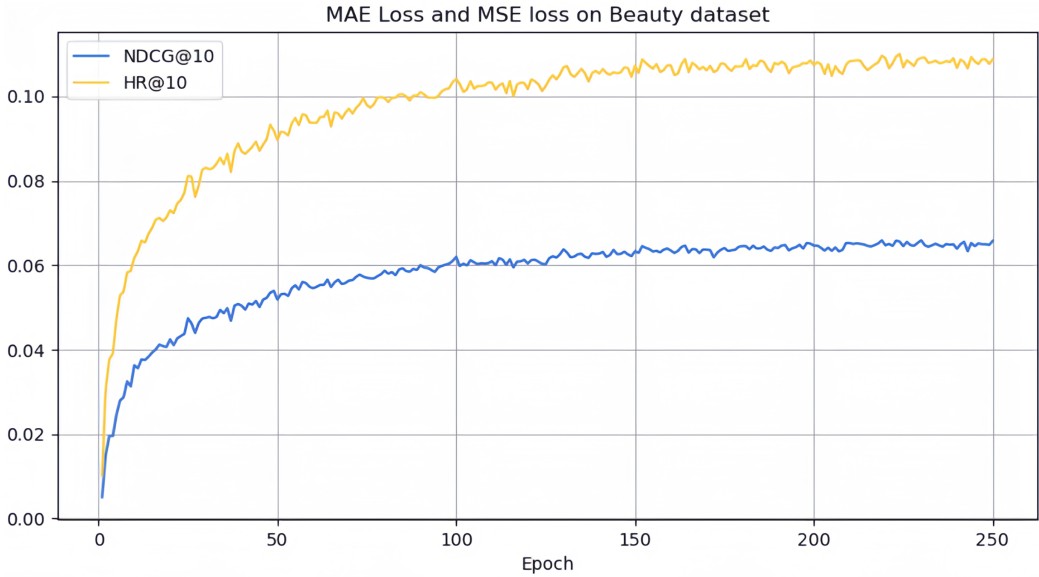

**Figure 4 NDCG@10 and HR@10 on Beauty dataset.**

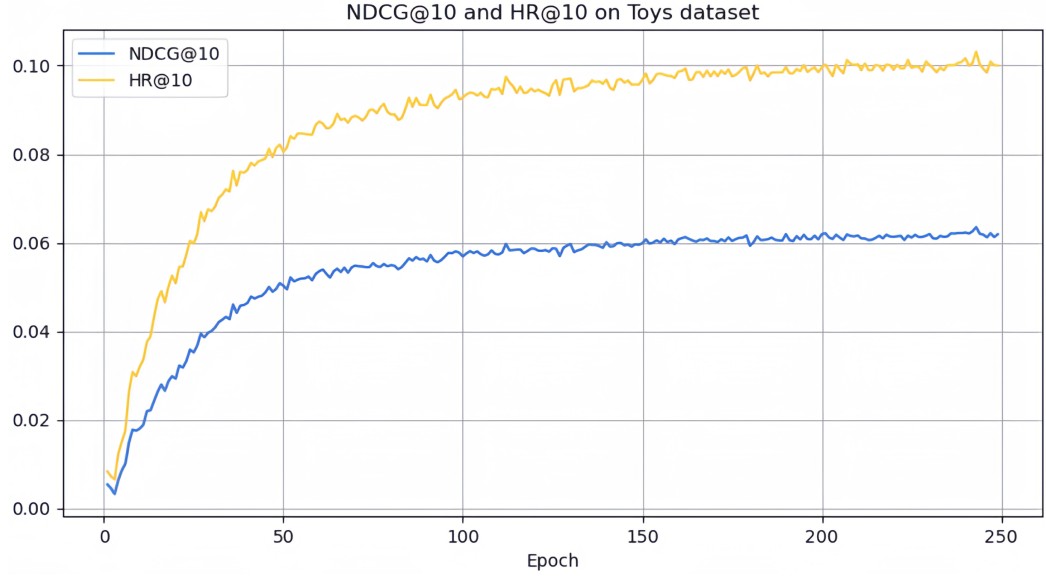

**Figure 5 NDCG@10 and HR@10 on Toys dataset.**

can be converted to using batch matrix multiplications. The second term in Eq. (7) can be transformed as a calculation of Euclidean norm as Eq. (21):

$$
\text{trace}\left(\sum_{S_t} + \sum_{S_k} - 2\left(\sum_{S_k}{}^{1/2}\sum_{S_t}\sum_{S_k}{}^{1/2}\right)^{1/2}\right) = \left\|\sum_{S_t}{}^{1/2} - \sum_{S_k}{}^{1/2}\right\|_F^2 \tag{21}
$$

where $\|\cdot\|_F^2$ is the Frobenius norm, which can be calculated by matrix multiplications. Also, since $\sum_{S_t}$ and $\sum_{S_k}$ are both diagonal matrices, we can further reduce the computational complexity to: $\frac{nd}{2} + \frac{n^2 d}{2} + 2n^2$, The Euclidean norm of the mean

**Peer**J Computer Science

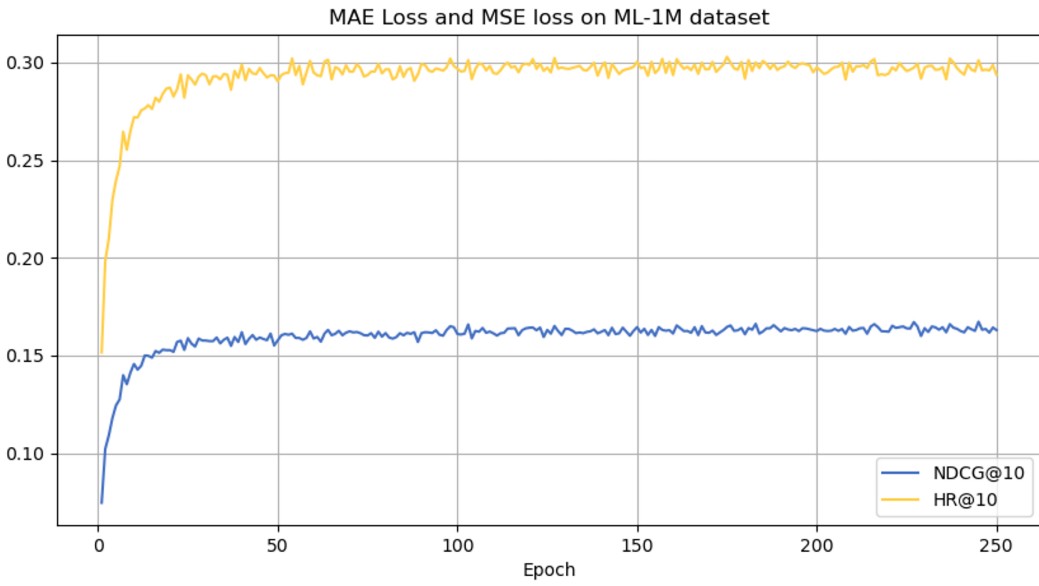

**Figure 6 NDCG@10 and HR@10 on ML-1M dataset.**

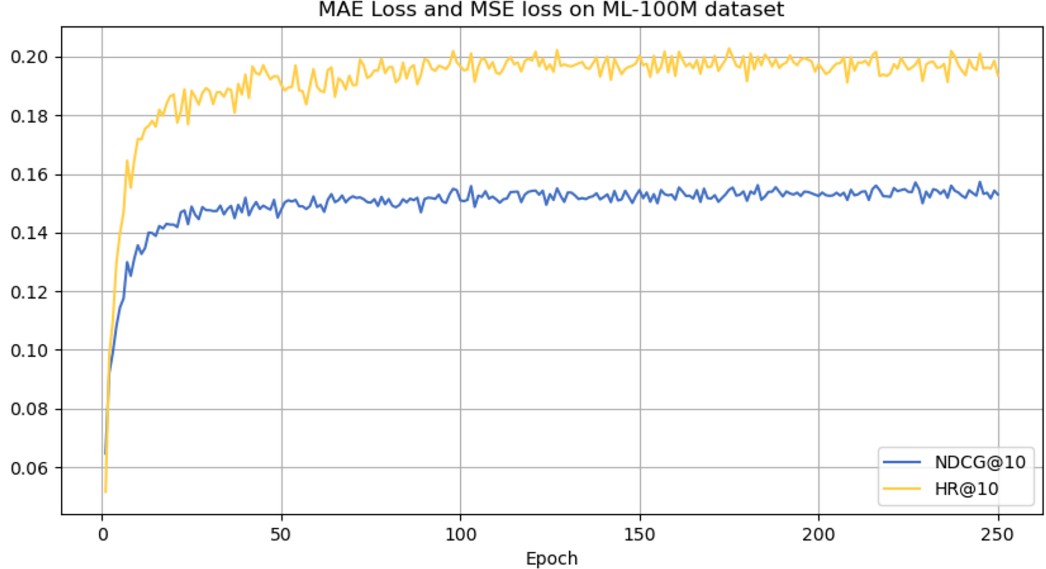

**Figure 7 NDCG@10 and HR@10 on ML-100M dataset.**

embeddings part in Eq. (7) can also be calculated by matrix multiplications with the same time complexity. Therefore, the overall time complexity of the Wasserstein self-attention is: $O(nd + n^2d + 4n^2)$. The random cloze task is a mechanism for generating positive and negative samples by randomly masking part of the sequence. Assuming that each sample generates $P$ negative samples, the overall complexity of the random cloze task is $O(B \cdot P \cdot n)$, where $B$ is the batch size. In contrastive learning, the main computational

**Table 3 The average running time of each epoch.**

|  | Beauty (s) | Toys (s) | ML-1M (s) | ML-100M (s) |
|---|---|---|---|---|
| HGN | 2.063 | 2.281 | 1.852 | 2.764 |
| CoSeRec | 1.879 | 1.988 | 1.297 | 2.657 |
| LightGCN | 1.894 | 1.959 | 1.583 | 1.986 |
| TGT | 0.957 | 1.392 | 0.795 | 1.312 |
| HAM | 0.820 | 0.991 | 0.701 | 1.178 |
| CBiT | 0.883 | 0.892 | 0.315 | 1.031 |
| MADGA | 0.796 | 0.601 | 0.219 | 0.642 |
| SwiMDiff | 0.561 | 0.653 | 0.375 | 0.523 |
| CLWSR | 0.216 | 0.640 | 0.085 | 0.461 |

complexity comes from the generation of positive and negative samples, the similarity calculation, and the loss function calculation. Therefore, the general complexity of contrastive learning is $O(B \cdot P \cdot n \cdot d + B \cdot P)$. By also considering the feed-forward networks, we obtain the final asymptotic computational complexity as $O\left(nd + n^2d + 4n^2 + \frac{nd^2}{2} + B \cdot P \cdot n \cdot (1 + d) + B \cdot P\right)$. The computational complexity of traditional self-attention is $O(n^2d + nd^2)$. Note that both complexities are typically dominated by the $O(n^2d)$ term as $d$ is typically much larger than 4. This indicates that CLWSR has asymptotic time complexity similar to general conventional models.

We use the same GPU to measure the average running time per epoch. We compare the running time of the proposed model with several benchmarks. As shown in Table 3, CLWSR achieves the best results on Beauty, ML-1M, and ML-100M. The running time on the Toys datasets is moderate. Further analysis of the dataset in Table 2 shows that CLWSR performs better when operating on datasets with longer average sequence lengths, as Beauty, ML-1M, and ML-100M have relatively longer average sequences, so they effectively capture dependency relationships at various positions within extended sequences.

## Ablation studies

We perform ablation studies of augmentation strategies that utilize stochastic embedding and Wasserstein self-attention mechanism to evaluate their effectiveness in this section.

We use a stochastic Gaussian distribution for each item, leveraging covariance to introduce sequence uncertainty into our model. The development of a novel Wasserstein self-attention model aids in defining item-item positional relationships within sequences, thereby effectively integrating uncertainty into the model training process. Moreover, the Wasserstein self-attention mechanism encourages collaborative and objective learning by adhering to the triangle inequality principle. Our model incorporates both stochastic embedding and Wasserstein self-attention mechanisms.

To evaluate these two strategies' individual efficacy, we conducted ablation experiments on our approach, modifying only one of the mechanisms while maintaining optimal

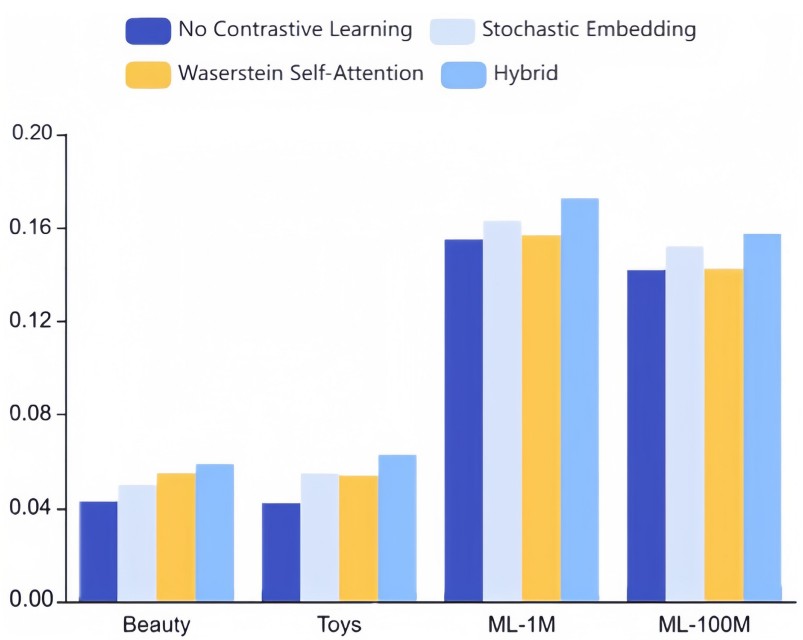

**Figure 8 Ablation studies on stochastic embedding and Wasserstein self-attention mechanism for augmentation strategies (NDCG@10).**

settings for the other hyperparameters. As depicted in Fig. 8, enabling contrastive learning surpasses a model without contrastive learning altogether. Notably, neither stochastic embedding nor the Wasserstein self-attention mechanism in isolation can achieve performance comparable to our combined hybrid strategy. This disparity can be traced to two key factors:

1. The amalgamation of these mechanisms introduces additional perturbations to the original sequence, yielding more challenging and higher-quality samples for comparative learning.

2. The collaborative and objective learning ingrained within CLWRS, where such principles aid cold items in identifying more collaborative neighbors, as popular items may exhibit correlations through triangular inequalities. This underlines the collaborative transferability signals' requisite and superiority in the realm of sequential recommendation.

## Discussion

The primary distinction between Wasserstein self-attention and the conventional attention mechanism lies in the calculation of attention weights. While the traditional attention mechanism employs dot product, Wasserstein self-attention utilizes negative 2-Wasserstein distances. As a result, the distribution of attention weights in Wasserstein self-attention tends to be more uniform compared to the traditional attention mechanism, which tends to focus on a select few items in the sequence. This difference can be attributed to the concept of collaborative transmissibility, which facilitates a closer association between neighboring items and introduces a wider range of collaborative neighbors in the modeling of item-item transitions.

Although the CLWSR model has achieved good performance on datasets of different sizes and its time cost is also low, the model still has some limitations, mainly manifested in the performance bottleneck on datasets with short sequence. For example, on the Beauty and Toys datasets, whose average sequence lengths are 8.9 and 8.6, respectively, the CLWSR model is not as good as the MADGA model in terms of NDCG. The reason is that the model we proposed combines the cloze task mask and dropout mask mechanisms to generate high-quality positive samples, but the sequence is short, and the cloze task mask and dropout mask mechanisms are difficult to play a role, which affects the performance of the model to a certain extent.

## CONCLUSIONS

In this article, a pioneering Wasserstein self-attention sequential recommendation model is introduced, employing a multidimensional elliptical Gaussian distribution to represent items. The elliptical Gaussian distribution consists of a vector of means and a vector of covariances, where the covariance factor reflects the underlying item uncertainty. This framework is utilized to model dynamic uncertainties and capture synergistic transferability within the recommendation system. Additionally, a novel regularized BPR loss function is introduced to ensure a considerable separation between positively and negatively sampled items. The incorporation of the cloze mask task and dropout mask enables the generation of multiple positive samples, expanding the scope of pairwise comparison learning to encompass multiple pairs of instances. We conducted experiments on four public datasets, and the results show that our proposed method outperforms several contemporary methods, underscoring the effectiveness of our strategy in tackling the cold start item recommendation challenge and emphasizing the significance of collaborative transferability in sequential recommendation tasks.

### Funding

This work was supported by Science and Technology Research Projects of Henan Province, China (242102211033). There was no additional external funding received for this study. The funders had no role in study design, data collection and analysis, decision to publish, or preparation of the manuscript.

### Grant Disclosures

The following grant information was disclosed by the authors:
Science and Technology Research Projects of Henan Province, China: 242102211033.

### Competing Interests

Shengbin Liang is an associate professor at Henan University. Jinfeng Ma, Tingting Chen, Xixi Lu, Shuanglong Ren are condidate masters at Henan University. Chenyang Zhao is an undergraduate student at Henan University. Qiuchen Zhao is an employee of Shandong Jining Tobacco Co. Lei Fu is an employee of China National Tobacco Corporation

Shandong Province Company and Huichao Ding is an employee of Nanyang Radio and Television Station.

## Author Contributions

- Shengbin Liang conceived and designed the experiments, prepared figures and/or tables, authored or reviewed drafts of the article, and approved the final draft.
- Jinfeng Ma conceived and designed the experiments, performed the experiments, analyzed the data, performed the computation work, prepared figures and/or tables, authored or reviewed drafts of the article, and approved the final draft.
- Qiuchen Zhao conceived and designed the experiments, performed the experiments, analyzed the data, prepared figures and/or tables, authored or reviewed drafts of the article, and approved the final draft.
- Tingting Chen conceived and designed the experiments, performed the experiments, analyzed the data, performed the computation work, prepared figures and/or tables, authored or reviewed drafts of the article, and approved the final draft.
- Xixi Lu conceived and designed the experiments, performed the experiments, analyzed the data, prepared figures and/or tables, authored or reviewed drafts of the article, and approved the final draft.
- Shuanglong Ren conceived and designed the experiments, performed the experiments, analyzed the data, prepared figures and/or tables, authored or reviewed drafts of the article, and approved the final draft.
- Chenyang Zhao conceived and designed the experiments, performed the experiments, analyzed the data, performed the computation work, prepared figures and/or tables, authored or reviewed drafts of the article, and approved the final draft.
- Lei Fu conceived and designed the experiments, performed the experiments, analyzed the data, prepared figures and/or tables, authored or reviewed drafts of the article, and approved the final draft.
- Huichao Ding performed the experiments, analyzed the data, performed the computation work, prepared figures and/or tables, authored or reviewed drafts of the article, and approved the final draft.

## Data Availability

The third party data is available at:

- ML-1M and ML-100M: https://grouplens.org/datasets/movielens.
- Beauty: https://www.kaggle.com/datasets/skillsmuggler/amazon-ratings.
- Toys: https://www.kaggle.com/datasets/PromptCloudHQ/toy-products-on-amazon/data.

## Supplemental Information

Supplemental information for this article can be found online at http://dx.doi.org/10.7717/peerj-cs.2749#supplemental-information.

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
