# Peer review of "A sequential recommendation method using contrastive learning and Wasserstein self-attention mechanism"

_PeerJ Computer Science, doi:10.7717/peerj-cs.2749_

## Round 0.1 · original submission · Major Revisions

Dear Authors,

please check carefully the suggestions given by the reviewers and address ALL their comments. In particular the paper is very difficult to read and the novelty should be better clarified.

Best regards,

M.P.

Reviewer 1 ·

Basic reporting

This research paper proposes a contrastive learning based sequential recommendation system with attention mechanism. The paper is reasonably well organized. However, the related work section could benefit from a more in-depth comparison with recent transformer-based and graph neural network-based approaches to sequential recommendation. This would clarify how the proposed method contrasts with or improves upon these methods.

Experimental design

The novelty of the approach may be clearly stated in contrast to the base model CBiT (Fan, Z., Liu, Z., Wang, Y., Wang, A., Nazari, Z., Zheng, L., ... & Yu, P. S. (2022, April). Sequential recommendation via stochastic self-attention. In Proceedings of the ACM web conference 2022 (pp. 2036-2047).

Validity of the findings

The performance of the proposed approach is compared with 6 other research works and has shown increment in the evaluation metrics.
The authors may also refer the following research work which seems to present better performance on Beauty dataset:
Wang, C., Ma, W., Chen, C., Zhang, M., Liu, Y., & Ma, S. (2023). Sequential recommendation with multiple contrast signals. ACM Transactions on Information Systems, 41(1), 1-27.

A comparative analysis with recent similar approaches is also recommended.

Additional comments

This paper addresses a timely and relevant problem in sequential recommendation.
While the paper is well organized, the novelty and contributions need to be clearly stated.
The paper would also benefit from more detailed comparison with baseline models in different domains.
Discussing limitations like model complexity or training time on large datasets may be included
The quality of the figures and equations needs improvement

Reviewer 2 ·

Basic reporting

This paper introduces a new method to improve recommendation systems by addressing two main challenges: unpredictable user behavior and limitations in existing techniques for capturing relationships between items. Traditional methods struggle with these challenges, especially in real-world scenarios.

The authors propose using random self-attention with a Gaussian distribution to handle unpredictability, modeling item relationships more accurately. They also introduce a novel contrastive learning approach that creates better training samples and a balanced loss function to improve results.

In experiments, the model outperformed top-performing methods, particularly with difficult cases like "cold start" items (new or less-seen items). It achieved notable improvements in recommendation accuracy across three datasets, with up to a 68% increase in key performance metrics on one dataset.

Experimental design

The authors conducted experiments using three well-known datasets: Beauty and Toys (both from the Amazon dataset) and MovieLens-1M (ML-1M). The Beauty and Toys datasets contain product reviews across various domains with relatively short user interaction sequences, while ML-1M consists of movie ratings and has longer user interaction sequences.

To prepare the data, they followed established preprocessing techniques, removing duplicate interactions and organizing each user's interactions in chronological order. They also filtered out users with fewer than five interactions and items that were rated by fewer than five users. The evaluation followed a leave-one-out setup, where the last interaction in each user sequence was set aside for testing, the second-to-last for validation, and the rest for training. Table 1 in the paper provides a statistical summary of the processed datasets.

This rigorous experimental setup, along with the careful preprocessing, strengthens the reliability of their findings across diverse types of user-item interactions.

Validity of the findings

The proposed model, CLWSR, was tested against multiple established baselines, including CoSeRec, CBiT, HGN, LightGCN, TGT, and HAM, each bringing unique methods for sequential recommendation tasks. Results show that CLWSR consistently outperformed these baselines across three datasets—Beauty, Toys, and ML-1M—with particularly strong performance on the ML-1M dataset, which has longer user interaction sequences. This improvement is largely attributed to the model's Wasserstein self-attention mechanism, which effectively captures dependencies within these longer sequences.

Though CLWSR slightly trails HAM on one specific metric (HR@5) in the Beauty dataset, it excels overall due to its integration of distributed embedding, Wasserstein self-attention, and multi-pair comparison learning. Performance improvements were confirmed through ablation studies, which revealed that using either stochastic embedding or Wasserstein self-attention in isolation was less effective than combining both. Also, CLWSR’s use of the Wasserstein self-attention mechanism encourages collaborative transfer, benefiting “cold start” items by connecting them with similar items. This approach, by introducing uncertainty through a Gaussian distribution and leveraging multi-pair contrastive learning, consistently generates higher-quality results across all datasets.

Additional comments

While the proposed model demonstrates impressive results, additional analysis comparing its performance with other self-attention methods could strengthen the findings. A more detailed discussion on how Wasserstein self-attention specifically differs from traditional self-attention models—particularly in terms of handling long-term dependencies and uncertainty—would provide valuable insights. Also, outlining the limitations of the study would enhance transparency. For instance, the model’s potential computational complexity, scalability concerns for larger datasets, or performance in domains beyond the tested benchmarks could be addressed. This would help in identifying areas for improvement and guide future research directions.

·

Basic reporting

There are too many graphical mistakes in the paper that prevent a honset evaluation of the manuscript. Therefore the paper must be resubmitting correcting the graphical mistakes that appears in the mathematical symbols, for instance, at line 117, 121, 122, there are several reverse question marks that indicate that any symbol was not correctly displayed. The equations must be all enumerated, and the graphical quality of equation must be improved (please, use Latex !).
The structure of the paper should be improved. There are too many "widows", a chapter title, followed by the title of a subsection without any text.
In this form, the manuscript cannot be read. The authors must improve the graphical quality of the paper and of the equations. I suggest to rewrite the paper using Latex.

Experimental design

I don' t comment the experimental design, since the paper is very difficult to read (see Section 1)

Validity of the findings

I don' t comment this section, since the paper is very difficult to read (see Section 1)

Additional comments

I don' t comment this section, since the paper is very difficult to read (see Section 1)

---

## Round 0.2 · Minor Revisions

Some remaining minor concerns should be addressed.

Reviewer 1 ·

Basic reporting

No comment

Experimental design

In the 'conclusion' section the authors have mentioned "a novel regularized Bayesian Personalized Ranking (BPR) loss function is introduced to ensure a considerable separation between positively and negatively sampled items". However, the incorporation of regularized BPR is not mentioned in the previous sections/ formulation. What is the novelty?

Validity of the findings

No Comment

Additional comments

The paper has been reasonably well revised. The authors should incorporate regularized BPR in the methodology section

·

Basic reporting

The authors fully addressed the reviewers' comments and the manuscript can be published in the present form,

Experimental design

It is adequate.

Validity of the findings

The validity of the findings is good.

Additional comments

No further comments.

---

## Round 0.3 · accepted · Accept

Reviewers are satisfied with the revisions, and I concur to recommend accepting this manuscript.

Reviewer 1 ·

Basic reporting

no comment

Experimental design

no comment

Validity of the findings

no comment

Additional comments

The authors have incorporated the suggestions mentioned.